# LLM Hallucination Detection: A Fast Fourier Transform Method Based on Hidden Layer Temporal Signals

## Abstract

Hallucination remains a critical barrier for deploying large language models (LLMs) in reliability-sensitive applications. Existing detection methods largely fall into two categories: factuality checking, which is fundamentally constrained by external knowledge coverage, and static hidden-state analysis, that fails to capture deviations in reasoning dynamics. As a result, their effectiveness and robustness remain limited. We propose HSAD (Hidden Signal Analysis-based Detection), a novel hallucination detection framework that models the temporal dynamics of hidden representations during autoregressive generation. HSAD constructs hidden-layer signals by sampling activations across layers, applies Fast Fourier Transform (FFT) to obtain frequency-domain representations, and extracts the strongest non-DC frequency component as spectral features. Furthermore, by leveraging the autoregressive nature of LLMs, HSAD identifies optimal observation points for effective and reliable detection. Across multiple benchmarks, including TruthfulQA, HSAD achieves over 10 percentage points improvement compared to prior state-of-the-art methods. By integrating reasoning-process modeling with frequency-domain analysis, HSAD establishes a new paradigm for robust hallucination detection in LLMs.

## 1 Introduction

Large Language Models have demonstrated exceptional performance in tasks such as language understanding and code generation in recent years, establishing themselves as the foundational technology for various applications, including text generation, question-answering systems, and information extraction Touvron et al. (2023). However, the frequent occurrence of hallucinations during their generation process—defined as the production of results that are factually inconsistent or lack contextual support—not only undermines the credibility of LLM outputs but also severely restricts their deployment in high-stakes scenarios Ju et al. (2024); Li et al. (2024).

In cognitive neuroscience, a substantial body of experimental evidence indicates that when humans encounter scenarios involving information falsification or cognitive conflict, they exhibit specific psychological and neural signal changes over time. These include a gradual increase in cognitive load, fluctuations in attentional states, and the evolution of electroencephalogram signals in their spectral structure Lo & Tseng (2018). As illustrated in Figure 1, these time-evolving signal patterns can be regarded as observable physiological manifestations of internal conflict, which indirectly reflect an individual's cognitive state and the dynamic process of information processing. The analysis of such signals can provide a theoretical basis and technical support for the study of human deception detection. It has been empirically demonstrated that LLM often exhibit behavioral patterns similar to those of humans under circumstances of information fabrication. Therefore, by drawing inspiration from the signal modeling approaches in cognitive neuroscience, the LLM's reasoning process can be modeled as a temporal signal, offering a novel perspective for hallucination detection.

Inspired by this, the HSAD method models the hidden layer vectors from different stages of the LLM's reasoning process as a hidden layer temporal signal, analogous to the process of human deception detection, to perform hallucination detection on the LLM.

Specifically, as depicted in Figure 1, HSAD first extracts hidden layer vectors from the LLM's forward reasoning process and constructs them into a hidden layer temporal signal with temporal characteristics, ordered by layer. Subsequently, it employs the Fast Fourier Transform to map this temporal signal into the frequency domain, thereby constructing spectral features that reveal anomalous signals during reasoning. Experiments have demonstrated that these anomalous signals often correspond to hallucinations in the generated content. Building upon this foundation, the spectral features are further utilized as a discriminative basis to design a hallucination detection algorithm for identifying potential hallucinatory content within the LLM's generation process.

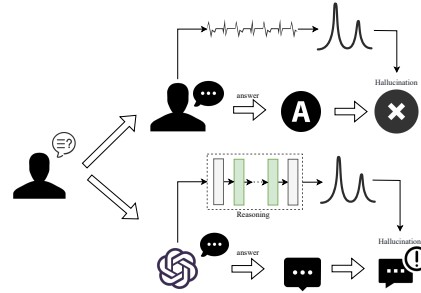

Figure 1: A comparison diagram of human lie detection vs. model's lie detection.

This strategy aims to characterize the evolutionary trajectory of the LLM's thought process in an interpretable manner, thereby enabling the effective detection of hallucinations Lieberum et al. (2024); Lindsey et al. (2025). Unlike most existing methods, HSAD does not rely on external knowledge bases for fact verification, but instead focuses on detecting anomalies within the LLM's internal signals. The main contributions of this paper are summarized as follows:

- We propose an LLM hallucination detection method, HSAD, which is analogous to the human deception detection mechanism. This method constructs a hidden layer temporal signal based on the LLM's inference process and introduces frequency-domain analysis and spectral feature construction to analyze and identify potential hallucinations. Theoretical derivations and experiments have demonstrated the feasibility and effectiveness of this approach.

- We design and implement an LLM hallucination detection algorithm based on spectral features. Specifically, we derive and determine the observation points for LLM hallucinations, then construct spectral features in the frequency domain for detection. This algorithm achieves SOTA performance on multiple standard datasets, while ablation experiments verify the individual contributions and synergistic gains of each module.

## 2 RELATED WORK

Existing hallucination detection methods largely build on LLM interpretability research, which can be grouped into two directions: (1) fact-consistency verification and (2) hidden representation analysis. Interpretability studies aim to uncover internal mechanisms and decision logic of LLMs, providing a foundation for improving reliability and controllability Ji et al. (2024). Representative approaches include observation-based methods (e.g., probing analysis, Logit lens, sparse representations, cross-model explanations), intervention-based methods (e.g., activation patching Zhao et al. (2024a)), hybrid methods, and other techniques such as transcoders Turpin et al. (2023), concept-driven explanations Wan et al. (2024), classifier injection, and SVD-based attention analysis. These methods offer theoretical support for identifying hallucination-related features and advancing systematic understanding of LLM behaviors.

**Fact-Consistency Verification.** This line of work aligns generated content with external authoritative knowledge to detect misinformation Zhao et al. (2024b). FActScore Kim et al. (2024) segments generations into atomic facts and verifies them against knowledge bases such as Wikipedia, but suffers from limited coverage and update frequency. SAFT Rawte et al. (2024) extends this idea with search-engine interactions, where LLM agents conduct multi-round verification to enhance factual evaluation of long texts, albeit at high computational cost and with the same knowledge coverage limitations.

**Hidden Representation Analysis.** Another direction analyzes internal representations to detect hallucinations Park et al. (2025). INSIDE Chen et al. (2024) parses dense semantic signals and applies feature clipping to regulate activations, reducing overconfidence. Duan et al. Greenblatt

et al. (2024) empirically showed distinct hidden states for truthful vs. fabricated responses. Probing-based studies further explore layerwise knowledge encoding Ju et al. (2024), concept specialization, and syntactic learning depth He et al. (2024). Recent methods also construct explicit latent spaces: TTPD Bürger et al. (2024) defines a "truth subspace" robust to negation and complex expressions, and TruthX Zhang et al. (2024) disentangles semantic and factual dimensions via autoencoding, editing hidden states to mitigate hallucinations.

# 3 MODELING AND ANALYSIS

To understand the internal cognitive mechanisms of LLM during the text generation process, this chapter establishes a formal mathematical model based on its forward-pass reasoning process and provides a criterion for hallucination discrimination. Subsequently, from the perspective of temporal modeling, the hidden layer vectors in different layers are regarded as the temporal unfolding of the reasoning process, thereby laying the foundation for subsequent frequency-domain modeling.

## 3.1 MODELING OF LLM INFERENCE PROCESSES

This section systematically elaborates on the computational logic and symbolic system of hidden vectors in each layer of the LLM.

On this basis, a unified representation of the inference process is proposed and illustrated through diagrams.

Let the input question be denoted as $Q = (t_0^Q, t_1^Q, \ldots, t_{m-1}^Q)$ and the corresponding reference answer as $A = (t_0^A, t_1^A, \ldots, t_{n-1}^A)$, where $t_i$ represents the $i$-th token in the input or output sequence. As illustrated in Figure 2, for an LLM with $l$ decoder layers, the computation at each layer can be expressed as follows.

The attention vector $ah_i^j$ of token $t_i$ at layer $j$ is given by Equation 1.

$$ah_i^j = ATT(h_i^{j-1}, k_{cache}, v_{cache}) \quad (1)$$

The MLP vector $mh_i^j$ of token $t_i$ at layer $j$ is given by Equation 2.

$$mh_i^j = MLP(rh_i^j) \quad (2)$$

The final hidden vector $h_i^j$ of token $t_i$ at layer $j$ is given by Equation 3.

$$h_i^j = (ATT \circ MLP)(h_i^{j-1}) \quad (3)$$

The processing of tokens $t_0, \ldots, t_i$ by the LLM is defined as the $i$-th inference step $F_i$, as shown in Equation 4.

$$F_i(t_0, \ldots, t_i) = (emb \circ Layer^1 \circ \cdots \circ Layer^l$$
$$\circ unemb)(t_0, \ldots, t_i) \quad (4)$$

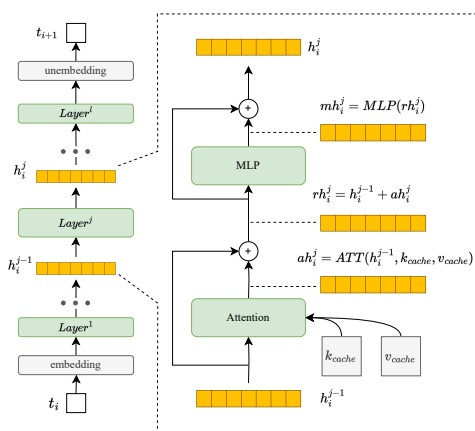

Figure 2: Schematic diagram of $F_i$

During the LLM inference process, the hidden vectors are continuously updated and integrated with the semantic context, exhibiting distinct temporal characteristics and information flow. Therefore, the entire inference process can be analogized to a thought process that evolves over time. Based on this, the hidden vectors from different layers are constructed into hidden layer temporal signals for modeling and analysis in the frequency domain.

## 3.2 LLM HIDDEN LAYER TEMPORAL SIGNAL CONSTRUCT

The LLM hidden layer temporal signal is defined as $X \in \mathbb{R}^{4l}$, which essentially represents a cross-layer sampling of hidden vectors, used to analyze the variation characteristics of the LLM's internal representations with increasing layer depth.

The specific construction process of the hidden layer temporal signal is as follows.

Assume the LLM has $l$ layers of Transformer architecture, with each layer containing an attention sub-layer and an MLP sub-layer, both with residual connections. From each layer, we sample the vectors of four key nodes corresponding to the generated token $t_{i+1}$, which are concatenated in the $j$-th layer to form a $d \times 4$ matrix $\mathbf{V}^j$, as shown in Equation 5.

$$\mathbf{V}^j = \begin{bmatrix} h_i^j \\ mh_i^j \\ rh_i^j \\ ah_i^j \end{bmatrix}^T \in \mathbb{R}^{d \times 4} \tag{5}$$

The four key node vectors are respectively: (1) The attention output vector $ah_i^j$: reflects the modeling result of the current layer's contextual dependencies; (2) The attention residual vector $rh_i^j$: superimposes previous layer information with the attention output; (3) The MLP output vector $mh_i^j$: embodies the local decision basis after nonlinear feature mapping; (4) The $j$-th layer output vector $h_i^j$: the fused result of all sub-module outputs in this layer.

Concatenate all $\mathbf{V}$ matrices from each layer $l$ to construct the matrix $\mathbf{T}$, as shown in Equation 6.

$$\mathbf{T} = [\mathbf{V}^l, \ldots, \mathbf{V}^j, \ldots, \mathbf{V}^1]^T \in \mathbb{R}^{4l \times d} \tag{6}$$

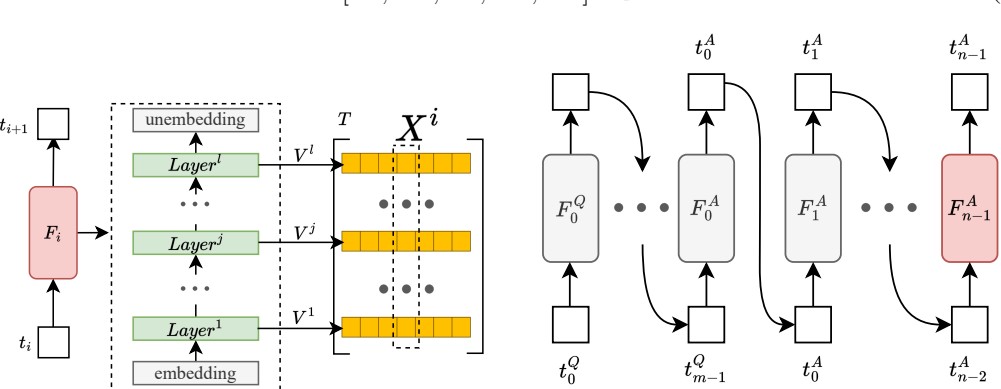

(a) Construction of hidden layer temporal signals      (b) $F_{n-1}^A$ global reasoning implication

Figure 3: Illustration of hidden signal construction and reasoning implication

As illustrated in Figure 3a, in matrix $\mathbf{T}$, each column corresponds to the temporal evolution of a specific dimension. For $i \in [1, d]$, the $i$-th column of matrix $\mathbf{T}$ is defined as the hidden layer temporal signal $X^i$ for the $i$-th dimension, which characterizes the variations of this dimension throughout the LLM inference process.

## 3.3 LLM HALLUCINATION DISCRIMINATION

To determine whether the output of an LLM constitutes a hallucination, the discrimination criterion shown in Equation 7 is employed.

$$K(A \mid Q) = \begin{cases} 1, & sim(A, A^*) \leq \tau \\ 0, & otherwise \end{cases} \tag{7}$$

Here, $Q$ represents the input question, $A$ denotes the response generated by the LLM, $A^*$ is the reference answer, $sim()$ indicates the semantic similarity score, and $\tau$ is the predetermined threshold. In our setting, the threshold is fixed at $\tau = 0.5$. Samples satisfying the above condition are classified as hallucinated outputs.

# 4 LLM HALLUCINATION DETECTION: A FAST FOURIER TRANSFORM METHOD BASED ON HIDDEN LAYER TEMPORAL SIGNALS

This chapter introduces a hallucination detection method based on the Fast Fourier Transform of hidden layer temporal signals. Initially, the hallucination observation points for the LLM are derived and identified. Subsequently, by applying the FFT to the hidden layer temporal signals, spectral features are constructed and their efficacy is demonstrated. Finally, hallucination detection is performed based on these spectral features.

## 4.1 DETERMINING OBSERVATION POINTS FOR LLM HALLUCINATIONS

In Section 3.1, the reasoning process $F_i$ of an LLM at each step has been formally defined. For an input question $Q$, its response $A = (t_0^A, \ldots, t_{n-1}^A)$ is generated incrementally over multiple steps, with the complete process corresponding to a series of reasoning states $F_0^A, \ldots, F_{n-1}^A$.

If one wishes to detect hallucinations based on internal signals, it is necessary to analyze the hidden layer temporal signals within these reasoning states. However, processing these states sequentially would incur significant computational costs. This leads to a critical question: is it necessary to analyze all reasoning states, or would a single representative node suffice.

From the perspective of the self-attention mechanism, the final-step reasoning state $F_{n-1}^A$ has already integrated the complete contextual information from the input $Q$ to the output $A$, and thus can serve as an observation point for the overall reasoning process.

In summary, as shown in Figure 3b, $F_{n-1}^A$ can be regarded as the global reasoning implication in the LLM input-output path, whose hidden layer vectors centrally reflect the LLM's understanding of the question $Q$ and its organizational capability for the answer $A$.

In layman's terms, $F_{n-1}^A$ represents the reasoning state of the model when generating the final token, at which point it has received the complete question $Q$ and all previously generated content, marking the moment of fullest understanding of the current task. This node not only centrally embodies semantic consistency and causal logic but also serves as an ideal observation point for detecting whether the LLM exhibits hallucinations.

## 4.2 CONSTRUCTION OF SPECTRAL FEATURES VIA FREQUENCY DOMAIN TRANSFORMATION OF HIDDEN LAYER TEMPORAL SIGNALS

In the field of signal processing, the analysis of signals often requires the simultaneous examination of both time-domain and frequency-domain characteristics. HSAD performs spectral analysis on the time signals of each dimension through the Fast Fourier Transform.

First, the Fast Fourier Transform as shown in Equation 8 is applied to each column of the hidden layer time-series signals $X^i$ in matrix $\mathbf{T}$.

$$Y_k^i = \sum_{n=0}^{N-1} X_n^i \cdot W_N^{nk} \tag{8}$$

Here, $Y_k^i$ represents the complex amplitude of the $k$-th frequency component after the transformation, $X_n^i$ denotes the $n$-th element of the $i$-th column in $\mathbf{T}$, $N = 4l$, and the rotation factor is given by Equation 9.

$$W_N^{nk} = e^{-j\frac{2\pi}{N}nk} \tag{9}$$

This factor maps the time-domain signal to the corresponding sine and cosine basis functions in the frequency domain, thereby revealing the strength of its components at different frequencies.

Empirical evidence shows that the DC component in the frequency domain typically corresponds to the overall offset of the input signal and is less indicative of variations during the reasoning process. Therefore, it is excluded during the feature extraction process.

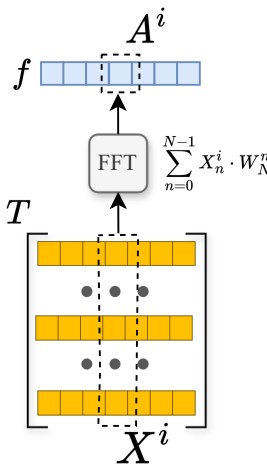

Figure 4: Spectral feature construction via FFT on built-in hidden layer temporal signals.

Specifically, only the magnitude of the largest non-DC frequency component is retained to capture temporal anomaly information in the LLM's reasoning process, as shown in Equation 10.

$$A^i = \max_{1 \leq k < N} |Y_k^i|, \quad i = 1, 2, \ldots, d \quad (10)$$

Finally, the frequency-domain representation vector $f$ is constructed, as shown in Equation 11.

$$f = [A^1, A^2, \ldots, A^d] \in \mathbb{R}^d \quad (11)$$

As shown in Figure 4, this frequency-domain vector $f$ represents the spectral features of the input $Q$, which can effectively characterize the anomalous perturbations in LLM during inference and provide structured input for hallucination detection.

### 4.3 HALLUCINATION DETECTION BASED ON SPECTRAL FEATURES

Based on the constructed spectral features, we build a hallucination detector $H : \mathbb{R}^d \to \{0, 1\}$. The input is the spectral feature $f$, and the output is the hallucination detection result.

For the question set $\text{Ques}(Q, A^*)$, $\forall Q$, the goal of the detector $H$ is to learn a binary predictor, as in Eq. 12.

$$H(Q) = \begin{cases} 1, & if\, K(A\,|\,Q) = 1 \\ 0, & otherwise \end{cases} \quad (12)$$

Here, $K(A\,|\,Q)$ indicates whether the answer $A$ is hallucinated.

To balance strong nonlinear modeling capacity with deployment efficiency, we adopt a MLP as the detector in the spectral-feature classification stage. The network consists of a feature extraction module and a classification module. The feature extraction module is composed of multiple fully connected layers, batch normalization, ReLU activation, and dropout, progressively compressing the feature dimensionality to 256. The classification module is a single fully connected layer that maps the 256-dimensional representation to the binary output space $\mathbb{R}^2$.

The core computation can be described as

$$h = ReLU\big(BN(Wf + b)\big), \quad (13)$$

$$\hat{y} = \sigma(Wh + b), \quad (14)$$

where $BN()$ denotes batch normalization and $\sigma()$ is the Sigmoid activation function, producing the hallucination probability $\hat{y} \in (0, 1)$. This design—staged dimensionality reduction coupled with regularization—captures complex spectral patterns without markedly increasing the parameter count and effectively suppresses overfitting.

During training, we use binary cross-entropy loss combined with $L_1$ regularization to control the sparsity of spectral-channel activations. The objective is given in Eq. 15:

$$\mathcal{L}_{\text{halluc}} = -\left[y \log \hat{y} + (1 - y) \log(1 - \hat{y})\right] + \lambda \|W_1\|_1, \quad (15)$$

where the first term encourages convergence to a reasonable decision boundary, and the $L_1$ regularizer enhances feature selectivity by attenuating non-discriminative frequencies, thereby improving generalization.

## 5 EXPERIMENTS

In this section, we first introduce the experimental setup and then present the advantages of HSAD compared with other hallucination detection methods across multiple datasets. Subsequently, we conduct detailed ablation studies to demonstrate the effects of frequency-domain modeling, cross-layer structures, key-node fusion, and the choice of appropriate hallucination observation points.

### 5.1 EXPERIMENTAL SETUP

This subsection describes the experimental setup in terms of datasets, evaluation models, evaluation metrics, baselines, and implementation details.

**Datasets.** We evaluate on four generative question answering (QA) tasks: three open-domain QA datasets—TruthfulQALin et al. (2022), TriviaQAJoshi et al. (2017), and NQ OpenKwiatkowski et al. (2019)—as well as one domain-specific QA dataset, SciQWelbl et al. (2017).

**Model types.** As shown in Table 1, two representative models are used for evaluation: LLaMA-3.1-8B and Qwen-2.5-7B-instruct.

**Base Methods.** As shown in Table 2, we compare our method with a comprehensive set of baseline approaches, including state-of-the-art methods Du et al. (2024). These baselines are categorized into four groups according to their underlying principles.

Table 1: Comparison of LLaMA-3.1-8B and Qwen-2.5-7B-instruct model architectures

| Feature | LLaMA-3.1-8B | Qwen-2.5-7B-instruct |
|---|---|---|
| Total Parameters | 8.0B | 7.61B |
| Number of Layers | 32 | 28 |
| Hidden Dimension | 4096 | 3584 |
| Context Length | 128k | 128k |
| Normalization | RMSNorm | RMSNorm |
| Released by | Meta AI | Alibaba |

Table 2: Categories of baseline methods

| Category | Methods |
|---|---|
| logit-based | Perplexity LN-entropy Semantic Entropy (SE) |
| consistency-based | Lexical Similarity (LS) SelfCKGPT EigenScore |
| verbalized | Verbalize Self-evaluation (Seval) |
| Internal-state-based | CCS HaloScope |

### 5.2 MAIN RESULTS

To validate the effectiveness and generalization ability of the proposed HSAD method, hallucination detection experiments were conducted on four public datasets: TruthfulQA, TriviaQA, SciQ, and NQ Open. HSAD was systematically compared with various existing detection methods. In the experiments, the BLEURT score Sellam et al. (2020) was used as the hallucination judgment threshold, and AUROC was adopted as the evaluation metric.

Table 3 reports the detection performance of different methods under various baseline models. It can be observed that HSAD consistently achieves the highest AUROC across all four datasets, with significant improvements over existing methods on multiple benchmarks.

In summary, HSAD demonstrates superior and stable detection performance across different datasets and baseline models, providing strong evidence of its practicality and cross-domain generalization capability for hallucination detection tasks.

### 5.3 MORE RESULTS

**Effectiveness of Frequency-domain Modeling.** To test whether spectral features provide essential discriminative information for hallucination detection, we replaced the Fast Fourier Transform module in the original HSAD with a direct use of hidden state time-series signals. In this variant, the maximum value of each hidden dimension was taken as the representation, while keeping all

Table 3: Hallucination detection performance of different methods on various datasets (AUROC, %)

| Model | Method | TruthfulQA | TriviaQA | SciQ | NQ Open |
|-------|--------|-----------|----------|------|---------|
| Qwen-2.5-7B-instruct | Perplexity | 65.1 | 50.2 | 53.4 | 51.2 |
| | LN-entropy | 66.7 | 51.1 | 52.4 | 54.3 |
| | SE | 66.1 | 58.7 | 65.9 | 65.3 |
| | LS | 49.0 | 63.1 | 62.2 | 61.2 |
| | SelfCKGPT | 61.7 | 61.3 | 58.6 | 63.4 |
| | Verbalize | 60.0 | 54.3 | 51.2 | 51.2 |
| | EigenScore | 53.7 | 62.3 | 63.2 | 57.4 |
| | Self-evaluation | 73.7 | 50.9 | 53.8 | 52.4 |
| | CCS | 67.9 | 53.0 | 51.9 | 51.2 |
| | HaloScope | 81.3 | 73.4 | 76.6 | 65.7 |
| | **HSAD** | **82.5** | **92.1** | **94.7** | **88.3** |
| LLaMA-3.1-8B | Perplexity | 71.4 | 76.3 | 52.6 | 50.3 |
| | LN-entropy | 62.5 | 55.8 | 57.6 | 52.7 |
| | SE | 59.4 | 68.7 | 68.2 | 60.7 |
| | LS | 49.1 | 71.0 | 61.0 | 60.9 |
| | SelfCKGPT | 57.0 | 80.2 | 67.9 | 60.0 |
| | Verbalize | 50.4 | 51.1 | 53.4 | 50.7 |
| | EigenScore | 45.3 | 69.1 | 59.6 | 56.7 |
| | Self-evaluation | 67.8 | 50.9 | 54.6 | 52.2 |
| | CCS | 66.4 | 60.1 | 77.1 | 62.6 |
| | HaloScope | 70.6 | 76.2 | 76.1 | 62.7 |
| | **HSAD** | **81.5** | **86.7** | **85.5** | **80.7** |

other settings identical. As shown in Fig. 5a and Fig. 5b, the full cross-layer modeling strategy significantly outperforms the other variants. This indicates that inter-layer dynamics encode essential reasoning features, which are critical for hallucination identification.

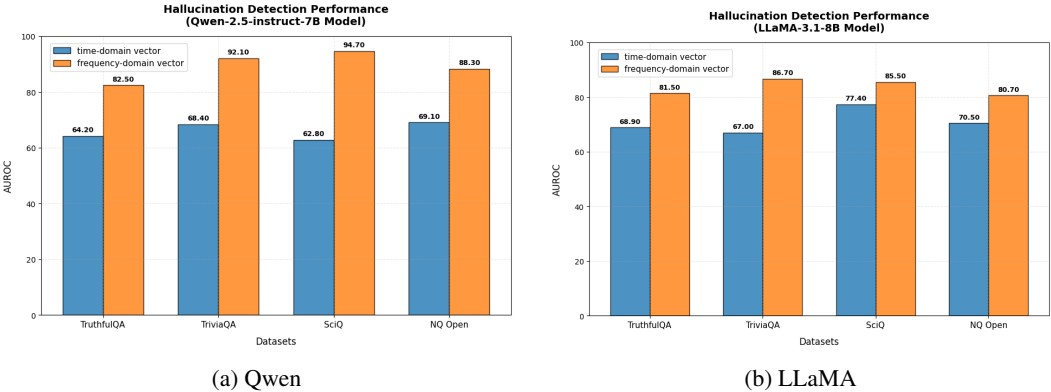

(a) Qwen         (b) LLaMA

Figure 5: Effect of frequency-domain modeling on detection performance

**Analysis of Observation Point Positions.** To assess the influence of different observation points on hallucination detection, we evaluated several positions: Q start, Q mid, Q end, A start, A mid, and A end. As illustrated in Fig. 6a and Fig. 6b, different observation points lead to significant differences in detection performance. In particular, A end (the end of the generated answer) achieves the best results, outperforming positions in the question segment (Q start/mid/end) and the earlier answer positions (A start, A mid).

**Cross-layer Structure Analysis.** To evaluate the impact of cross-layer modeling on LLM hallucination detection,We compared HSAD's performance under different sampling rates across datasets.

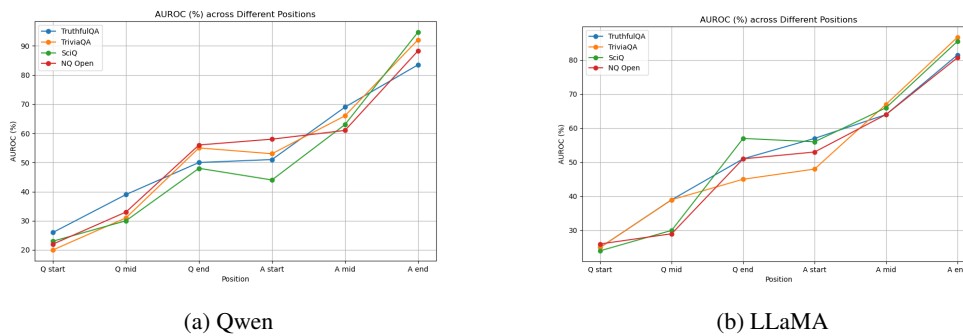

Figure 6: Comparison of observation point positions

Fig. 7a and Fig. 7b illustrate the relationship between the number of sampled layers and detection performance. As the number of sampled layers increases, performance steadily improves and saturates when all layers are included. This further verifies the integrative nature of spectral features along the layer dimension, highlighting the importance of treating the reasoning process as a holistic temporal cognitive system for modeling.

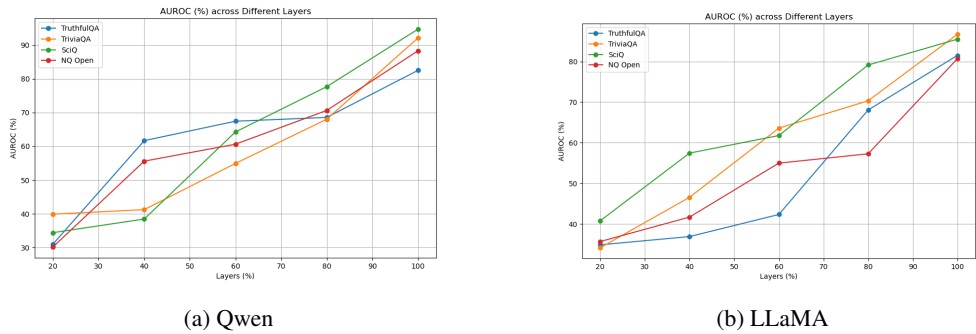

Figure 7: Effect of random layer sampling on performance

## 6 CONCLUSION

This study proposes a hallucination detection method for LLMs inspired by human polygraph mechanisms—HSAD (**H**idden **S**ignal **A**nalysis-based **D**etection). HSAD constructs hidden-layer time-series signals from the model's reasoning process and applies Fast Fourier Transform to capture frequency-domain patterns associated with hallucination behaviors. By leveraging spectral features, HSAD can detect abnormal hidden-state dynamics without relying on external knowledge, addressing key limitations of existing knowledge-dependent or behavior-only approaches. Rather than focusing solely on empirical performance, this work provides a new perspective for examining hidden-state behaviors of LLMs and offers a practical pathway toward implementing lightweight, model-internal safety mechanisms for high-reliability language generation.

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

## A  APPENDIX

Table 4: Performance Matrix(Qwen)

| Train and Test | NQ Open | SciQ | TruthfulQA | TriviaQA |
|---|---|---|---|---|
| NQ Open | 88.3 | 84.3 | 67.8 | 84.2 |
| SciQ | 77.7 | 94.7 | 77.1 | 82.2 |
| TruthfulQA | 74.9 | 80.7 | 82.5 | 79.5 |
| TriviaQA | 86.3 | 90.4 | 78.2 | 92.1 |

Table 4 reports the cross-dataset performance of our hallucination detector. Each row corresponds to the dataset used for training the detector, and each column corresponds to the dataset on which the detector is evaluated. Several observations emerge from the results.

These results are consistent with the hypothesis that hallucination behavior exhibits a relatively stable spectral structure across tasks, allowing HSAD-based detectors to generalize beyond their training domain.

