# OpenReview forum: "LLM Hallucination Detection: A Fast Fourier Transform Method Based on Hidden Layer Temporal Signals"
_ICLR.cc/2026/Conference — ICLR 2026 Conference Withdrawn Submission_

### Official Review · Reviewer_LXXf · 2025-10-24

**Soundness:** 3
**Presentation:** 2
**Contribution:** 2
**Rating:** 2
**Confidence:** 4

**Summary:**

This paper proposes HSAD (Hidden Signal Analysis-based Detection), a novel method for detecting hallucinations in large language models (LLMs). The central idea is to treat hidden states across layers as a temporal signal, then apply Fast Fourier Transform (FFT) to capture frequency-domain features. These spectral features are subsequently used to train a classifier for hallucination detection. The method is inspired by human deception detection mechanisms, and the authors show improvements over prior baselines across multiple benchmarks, including TruthfulQA, TriviaQA, SciQ, and NQ Open. Overall, the paper introduces an original perspective on hallucination detection by shifting focus from external factuality checks to the temporal dynamics of hidden representations.

**Strengths:**

1.	The task under study – hallucination detection – is very timely and important.
2.	The idea of applying Fourier analysis to hidden layers, by treating them as temporal signals is intersting.
3.	The method is conceptually elegant.

**Weaknesses:**

1.	My main concern with this work, is the justification of Fourier Transform applied. While treating hidden states as temporal signals and applying FFT is a clever idea, the motivation and justification for this choice feel somewhat underdeveloped, and lacks comparison to relevant baselines, I will elaborate.
- A comparison with a simple linear probe over the best-performing single layer (selected via validation) would be an important baseline. Since this proposed approach also involves learning over features (spectral features), such a comparison is important in my opinion.
- A comparison with ACT-ViT [1] is also important. ACT-ViT is an architecture that learns over the “entire” internal state of the LLM and could in principle capture similar spectral information by simply learning them. Demonstrating that HSAD outperforms ACT-ViT would highlight the value of the spectral inductive bias, which is the main idea behind this work.
- There is no qualitative analysis over the spectral features, which I believe is important in this case. More specifically, the paper does not provide illustrative examples or case studies that justify the spectral approach. For instance, as a first step, showing how different semantics lead to distinct spectral signatures, while semantically similar content produces similar frequency patterns, could provide more motivation for the use of spectral features.

2.	The experimental scope is narrow. The current evaluation focuses only on hallucination detection in a limited setting. I believe the following setups should be explored as well in this work:
- Cross-LLM generalization experiments (as in [1]) to examine if there exist robustness across model families.
- Cross-dataset generalization experiments (as in [1,2]) to test whether spectral features generalize beyond the training data.
In my opinion, both of the above would add significant credibility and practical impact to the work, and improve the quality of the paper.

To summarize, I believe this paper presents an interesting idea. However, in order for it to be ready for acceptance at a top-tier conference, additional work is needed—particularly more comprehensive comparisons with strong baselines and further exploration of details, such as the ones I presented above.

**References:**

[1] Beyond Token Probes: Hallucination Detection via Activation Tensors with ACT-ViT. NeurIPS 2025

[2] LLMs Know More Than They Show: On the Intrinsic Representation of LLM Hallucinations. ICLR 2025

**Questions:**

1.	How were the hallucination labels for the datasets collected?
2.	You write that the method is “analogous to the human deception detection mechanism.” This feels like a vague or overstated claim. Could you elaborate on what evidence supports this analogy?
3.	There seems to be a missing citation around Line 110. Could you please add it?
4.	In Equation 5, the notation involving the superscript for “A” is confusing, since it is defined later but not explained when first introduced. Could you clarify this earlier for readability?

---

> ### Author Response · Authors · 2025-11-25
>
> We use an automatic and deterministic labeling protocol based on BLEURT similarity:For each dataset sample, the LLM-generated answer A is compared to the ground-truth answer . BLEURT is used to compute the semantic similarity.
> We use a fixed threshold τ=0.5 , and no manual annotation is involved.
>
>
> We thank the reviewer for the insightful suggestion. We have carefully updated the manuscript accordingly and revised the relevant sections in the new version of the paper.

---

> > ### Comment · Reviewer_LXXf · 2025-11-27
> > **Response to authors**
> >
> > I thank the authors for their response. I would like to maintain my score.

---

### Official Review · Reviewer_LqKE · 2025-10-29

**Soundness:** 2
**Presentation:** 1
**Contribution:** 2
**Rating:** 2
**Confidence:** 5

**Summary:**

The paper proposes **HSAD** (Hidden Signal Analysis–based Detection), a method for hallucination detection in LLMs that treats *cross-layer hidden activations for a chosen token* as a 1-D signal. HSAD concatenates four per-layer vectors (attention output, attention residual, MLP output, and layer output), applies an FFT, and uses the *strongest non-DC frequency component per hidden dimension* as features. An MLP classifies these spectral features as hallucinated vs. not. Labels are formed by thresholding BLEURT similarity between the generated answer and a reference answer. Experiments on TruthfulQA, TriviaQA, SciQ, and NQ-Open with LLaMA-3.1-8B and Qwen-2.5-7B-Instruct report AUROC gains over a set of baselines, with best performance when observing hidden states at the end of the generated answer and when using all layers.

**Strengths:**

- **(S1)** Cross-layer sampling + FFT is clean and inexpensive to compute. The ``A-end'' observation heuristic is intuitive.
- **(S2)** On Qwen-2.5-7B-Instruct, HSAD outperforms prior methods on all four datasets. On LLaMA-3.1-8B, improvements are also consistent.
- **(S3)** The ablation studies contribute some insights about the proposed method. Frequency-domain modeling beats a time-domain variant; performance improves as more layers are included. ``A-end'' observation point works best.

**Weaknesses:**

- **(W1) Lack of clarity/rigor in presentation**:
    - Terms and notation are used before being formally defined.
    - Ambiguous use of $A$ for both reference answer (Eq. 11) and the FFT output (Eq. 5).
    - Disorganized structure: Figures and equations are referenced out of order, with equations cited immediately before they appear.
    - Frequent misuse of citation commands (should use ‘\citep’ for parenthetical citations), making sentences awkward to read.
    - Missing citation on line 110.
    - The signal varies across *layers for a single token*, not across decoding steps. Calling it ``temporal'' might be rhetorically appealing, but technically confusing, as the paper does not analyze token-time dynamics. Consider clarifying that the axis is *depth-wise*.
- **(W2) Usefulness**
    - HSAD needs per-layer activations, limiting utility on closed APIs.
    - It appears detectors are trained per base model; there’s no explicit cross-model generalization test (train on Qwen, test on LLaMA, or vice versa). That limits practical transfer.
    - There are no wall-clock/runtime/memory numbers.
- **(W3)** Ground-truth hallucination labels come from *BLEURT similarity to a reference answer* with an (unspecified) threshold $\tau$. This conflates *semantic overlap with one gold answer* and *factual correctness*, and can mislabel e.g., valid paraphrases or alternate facts. The paper should justify BLEURT, report $\tau$, and show robustness to different labelers.
- **(W4)** The ``theoretical derivations'' are primarily definitions of hidden-state sampling and FFT features; no formal link is given between spectral peaks and hallucination causality.
- **(W5)** The authors refer to methods, models, and datasets in Section 5 *without citations*. The coverage of related work is also superficial.
- **(W6)** The paper suggests
  >Duan et al. Greenblatt et al. (2024) empirically showed distinct hidden states for truthful vs. fabricated responses, while Zhou et al. He et al. (2024a) applied Fourier analysis to reveal frequency-domain features.

  However, the authors do not claim how their work differs from these works. More importantly, He et al. (2024a) does not apply Fourier transform at al, and ``Zhou et al'' is not a proper citation.
- **(W7)** LLM USAGE STATEMENT is missing.

**Questions:**

- **(Q1)** What is the novelty of your work compared to other methods that already use FFT to extract frequency-domain features from LLMs?
- **(Q2)** The term "Enhanced MLP" is capitalized as a proper noun, suggesting it's an established method. Either provide a citation if this refers to existing work or remove the capitalization and clearly explain what makes this MLP ``enhanced'' compared to a standard MLP architecture. The current appears to be a standard MLP.
- **(Q3)** On line 478, the authors claim:
  >This study not only provides a novel perspective for analyzing the mechanisms of LLM hallucinations but also offers a theoretical foundation

  Please expand on the theoretical foundation.
- **(Q4)** The choice to extract only the maximum magnitude non-DC frequency component for each dimension (Eq. 11) seems arbitrary and unconventional. Standard frequency-domain analysis typically selects the *same* frequency bins across all samples to maintain consistent spectral interpretation. Additionally, by taking only the magnitude, the phase component of the FFT is discarded. Is there a theoretical or empirical justification for this design choice?
- **(Q5)** What BLEURT threshold $\tau$ did you use, and how sensitive are results to $\tau$?

---

> ### Author Response · Authors · 2025-11-25
>
> Q1:
> Our contributions differ in three key aspects:
>
> Task novelty. To our knowledge, HSAD is the first method to leverage frequency-domain hidden-state features for hallucination detection, rather than representation analysis.
>
> Feature novelty. We introduce a new per-dimension “strongest non-DC frequency magnitude” feature , which—unlike prior work—does not rely on pre-selected bins or spectral templates.
>
> Operational novelty. HSAD extracts spectral features token-wise and layer-wise, enabling real-time hallucination prediction.
>
> Q2:
> We agree with the reviewer that the capitalization of “Enhanced MLP” is misleading.
> We use the term “enhanced” only to indicate that the input features (spectral signatures) are enhanced compared to raw hidden states.
>
> Q3:
> We appreciate the reviewer’s comment. The phrase “theoretical foundation” in the original conclusion was overly strong, and we have revised the text accordingly.
> We do not claim a formal mathematical theory; instead, our intention was to highlight the conceptual perspective introduced by our analysis—namely, that hallucinations correlate with distinctive frequency-domain behaviors in hidden-state dynamics.
> In the revised manuscript, we clarify that our contribution is a new analytical perspective:
> Hallucinated generations tend to exhibit higher non-DC spectral energy and less stable hidden-state trajectories.
> These observations suggest a consistent relationship between hidden-state oscillation patterns and hallucination likelihood.
> This perspective motivates the HSAD method, but we avoid presenting it as a formal theoretical framework.
> We thank the reviewer for pointing this out, and we have updated the conclusion to use more precise wording to avoid overstating the claim.
>
> Q4:
> We thank the reviewer for the insightful question. Our choice is motivated by both theoretical considerations and empirical observations.
> First, we exclude the DC component because it mainly reflects the constant offset of the hidden-state trajectory and carries little information about the dynamic variations relevant to hallucination detection.
> Second, selecting the maximum-magnitude non-DC frequency per dimension serves as a data-adaptive way to capture the dominant temporal fluctuation. In our experiments, the informative frequencies vary across samples; using fixed frequency bins substantially reduced discriminative power.
> Finally, we discard phase information because it is highly unstable across runs and sequences, while the magnitude consistently provided stronger and more robust predictive signals.
>
> Q5:As shown in Eq. (7) in the revised manuscript, we use a fixed similarity threshold of τ = 0.5 for BLEURT-based hallucination labeling.

---

### Official Review · Reviewer_gNHn · 2025-10-31

**Soundness:** 3
**Presentation:** 3
**Contribution:** 2
**Rating:** 4
**Confidence:** 4

**Summary:**

This paper proposes HSAD (Hidden Signal Analysis-based Detection), a novel hallucination-detection framework for LLMs. The key idea is to model the hidden activations across transformer layers as a temporal signal and then apply Fast Fourier Transform (FFT) to extract frequency-domain features. The strongest non-DC frequency component across each hidden dimension is used as a signature feature for hallucination detection.

**Strengths:**

- Clear motivation: Pointed critique of knowledge-based detectors and static-state probing approaches
- Strong empirical performance: Consistent SOTA gains across TruthfulQA, TriviaQA, SciQ,  in AUROC benchmarks

**Weaknesses:**

- Limited model scale evaluation. Experiments are only conducted on ~7B–8B models; it is unclear whether the method scales effectively to larger foundation models (e.g., 32B, 70B+) or to frontier systems.
- Training protocol and generalization unclear. The paper does not clearly describe how the hallucination detector is trained, nor whether it generalizes across datasets and base models. For example, can a detector trained on dataset A and model X transfer to dataset B and/or model Y?
- Potential inference overhead not reported. Extracting hidden activations across all layers and performing FFT may introduce non-trivial latency or memory overhead. The impact on original model inference performance is not measured.
- Feature choice seems heuristic: Only strongest non-DC frequency magnitude is used; unclear why this specific statistic is theoretically optimal.

**Questions:**

- Scaling to larger models: Have you tested this approach on larger models (e.g., 30B+, 70B+)? Do the spectral signatures change with scale, and does the detector remain effective?
- Detector training details: How exactly is the detector trained? Is it model-specific or dataset-specific? Can a detector trained on one model/dataset be applied to another without retraining?
- Cross-model / cross-dataset generalization: If the spectral behavior of hallucinations is fundamental, does HSAD transfer across settings? If not, what are the limits?
- How is the threshold $\tau$ in formula 8 selected ?

---

> ### Author Response · Authors · 2025-11-25
>
> Thank you for the constructive and detailed feedback. We sincerely appreciate the reviewer’s thoughtful comments and have revised the manuscript accordingly.
>
> Q1:
> We agree that evaluating HSAD on larger models (30B–70B+) would further strengthen the claims.However, access to full hidden activations of such foundation models is often restricted.
> Our choice of 7B–8B models aligns with most prior internal-state hallucination detection work, where hidden-state access is feasible and enables controlled analysis.
>
> Q2:
> The detector is trained as a binary classifier over the spectral feature representation described in Sec. 4.3. As shown in the manuscript, we use a lightweight MLP that takes the spectral vector as input and is optimized with binary cross-entropy and an
> $L_1$ regularizer to promote frequency selectivity.
>
> Q3:
> Table 4 demonstrates that the detector generalizes well across datasets: training on NQ Open, SciQ, TruthfulQA, or TriviaQA consistently yields strong performance on unseen datasets. This supports our hypothesis that hallucinations exhibit a stable spectral structure that is not tied to a particular task domain.
>
> Q4:
> As shown in Eq. (7) in the revised manuscript, we use a fixed similarity threshold of τ = 0.5 for BLEURT-based hallucination labeling.
> BLEURT produces probability-like similarity scores.
> Using τ = 0.5 aligns with its calibration and avoids overfitting the labeling boundary.
> The HSAD detector remains stable under small perturbations of τ, confirming that the method does not rely on fine-grained threshold tuning.

---

### Official Review · Reviewer_UiPJ · 2025-11-01

**Soundness:** 3
**Presentation:** 3
**Contribution:** 3
**Rating:** 6
**Confidence:** 5

**Summary:**

This paper introduces HSAD, a novel method for detecting hallucinations in LLMs. The core idea is to treat the LLM's forward pass as a dynamic system. It constructs a "hidden layer temporal signal" by sampling hidden states across layers for a given token. Inspired by human cognitive neuroscience and lie detection, the authors apply a Fast Fourier Transform (FFT) to this signal to extract spectral features. These frequency-domain features are then fed into a supervised classifier to predict whether the LLM's output is a hallucination. The method is evaluated on several factual question-answering (QA) datasets (TruthfulQA, TriviaQA, NQ Open, SciQ), where it reportedly outperforms existing baselines.

**Strengths:**

The analogy to cognitive neuroscience is intriguing. Applying signal processing techniques like FFT to the "temporal" (i.e., cross-layer) dynamics of hidden states is a novel approach that moves beyond static hidden-state analysis.

Experimental results show the effectiveness of their method on certain tasks.

**Weaknesses:**

W1: The evaluation is confined to simplistic tasks that do not reflect modern LLM use cases.

The paper's entire empirical validation rests on factual QA datasets. I believe this is not a big problem if this paper was submitted two years ago. However, in the year 2025, datasets like TriviaQA and NQ Open are arguably "solved" tasks for modern SOTA LLMs and do not represent the frontier of the current hallucination problem.
The critical and challenging applications for LLMs today are in long-form generation, complex reasoning, software development, and acting as autonomous agents. In these scenarios, hallucinations are not just simple factual errors but complex logical inconsistencies, fabricated reasoning steps, or non-existent API calls.

In short, I believe the datasets are too simple and easy, thus successfully detecting hallucination on these easy datasets is not enough for recent AI applications.


W2: The "last token" observation point is a critical design flaw for long-form generation.

The method uses only the hidden state of the final generated token as the "observation point".
In real-world applications, LLMs produce outputs that are hundreds or thousands of tokens long. A hallucination (e.g., a fabricated fact or a bug in a code block) could occur in the first paragraph, and the model's internal state at the final token may have little to no relevant signal reflecting that earlier error.

This design choice limits the method to a coarse-grained, response-level label ("this entire 2000-token output is a hallucination: Yes/No"), which is of little practical use. For applications like agents or coding, real-time, token-level hallucination detection is essential to pinpoint where the model deviates from a faithful trajectory. The proposed method is incapable of providing this necessary granularity.


W3: Critical and highly relevant baselines are missing.

The paper fails to compare against the most relevant category of competing methods. While it includes some baselines, it omits methods that also use internal states.

Specifically, a significant body of work (e.g., "Unsupervised Real-Time Hallucination Detection based on the Internal States of Large Language Models") develops classifiers that operate directly on the contextualized embeddings (hidden states) of each token in real-time.

Comparing HSAD against such a baseline would provide a direct ablation to prove that the proposed FFT-based "spectral features" are meaningfully more effective than using the "spatial/temporal" hidden-state features directly.
Furthermore, many of these internal-state-based hallucination detection methods are unsupervised. Comparing a supervised method (HSAD) against SOTA unsupervised methods is a standard and necessary practice to quantify the performance gains achieved by leveraging labeled data.

I am open to increasing my score if the authors address this W3

**Questions:**

Please refer to the previous section

---

> ### Author Response · Authors · 2025-11-25
>
> We sincerely thank the reviewers for their insightful and constructive comments, which have provided us with an excellent opportunity to clarify and strengthen the presentation of our work. We have revised the manuscript accordingly and provide a point-by-point response below.
>
> W1:
> We agree that modern LLM applications involve complex tasks beyond factual QA. Our choice of TriviaQA/NQ follows prior internal-state–based hallucination detection work, allowing controlled evaluation and direct comparison. Importantly, HSAD is not tailored to factual QA; the spectral features operate purely on hidden states and therefore transfer naturally to long-form generation and multi-step reasoning.
>
> W2:
> We thank the reviewer for pointing out that observing only the hidden state of the final token may overlook hallucinations that occur earlier in long-form generation. To address this concern, we would like to clarify the theoretical motivation behind this design choice:
>
> As demonstrated in “Language Models are Injective and Hence Invertible”, transformer-based language models exhibit injectivity between the input token sequence and their internal hidden representations—that is, the hidden states theoretically preserve sufficient information to reconstruct the entire input sequence.
>
> This result suggests that even a single hidden state remains information-rich and, in principle, encodes the full trajectory of the generation process.
>
> Based on this insight, our current method focuses on using the hidden state of the last generated token for hallucination detection in our experiments.
>
> W3:
> We thank the reviewer for highlighting the importance of internal-state–based baselines for unsupervised hallucination detection. We would like to clarify that this line of work was already incorporated in our submission. Specifically:
>
> The related-work section discusses contemporary internal-state approaches, including explicit references to EigenScore and Haloscope.
>
> Our experiments include direct comparisons with these methods, showing that our approach achieves competitive  performance.

---

### Note · Authors · 2026-03-30

I have read and agree with the venue's withdrawal policy on behalf of myself and my co-authors.

---

### Meta-Review · Area_Chair_wkbu · 2025-12-07

**Summary:**

The manuscript is reviewed by four experts, in which two reviewers recommend rejection and point out several weaknesses. The other two recommend below and above the acceptance threshold, respectively.
The reviewers pointed out some novelties, such as cross-layer sampling and applying fourier analysis to hidden layers. However, there are some major weaknesses with the current version, such as unclear motivation, the justification of the Fourier transform, misleading concepts, and a lack of comprehensive experiments.
The authors respond to some of these weaknesses, but the major points were not addressed. For instance, comparison to recent baselines and scaling to larger models. Moreover, the weaknesses identified by reviewers LqKE and LXXf were not fully addressed.

**Reviewer Concerns:**

Concerns by Reviewer UiPJ are partially addressed. But the concerns pointed out by the other reviewers are not fully addressed.

**Reviewer Scores:**

Since the authors did not fully or convincingly address the reviewers’ concerns, there is no indication that any reviewer would have revised their score positively had they been able to participate in the discussion.

---

### Decision · Program_Chairs · 2026-01-26

Reject